# Plasmonic gain in current biased tilted Dirac nodes

Sang Hyun Park[1], Michael Sammon[1], Eugene Mele[2] & Tony Low [1] ✉

Surface plasmons, which allow tight confinement of light, suffer from high intrinsic electronic losses. It has been shown that stimulated emission from excited electrons can transfer energy to plasmons and compensate for the high intrinsic losses. To-date, these realizations have relied on introducing an external gain media coupled to the surface plasmon. Here, we propose that plasmons in two-dimensional materials with closely located electron and hole Fermi pockets can be amplified, when an electrical current bias is applied along the displaced electron-hole pockets, without the need for an external gain media. As a prototypical example, we consider $WTe_2$ from the family of 1T′-$MX_2$ materials, whose electronic structure can be described within a type-II tilted massive Dirac model. We find that the nonlocal plasmonic response experiences prominent gain for experimentally accessible currents on the order of mA$\mu$m$^{-1}$. Furthermore, the group velocity of the plasmon found from the isofrequency curves imply that the amplified plasmons are highly collimated along a direction perpendicular to the Dirac node tilt when the electrical current is applied along it.

Surface plasmons[1] are collective excitations of electrons that result in highly confined electromagnetic modes in metals or semiconductors. In two-dimensional materials[2,3], such as graphene[4,5], propagating surface plasmons have been observed[6–8] and are accompanied by a variety of interesting phenomena such as plasmonic waveguiding[9,10], topological plasmons[11–13], chiral directional plasmons[14,15], and the plasmonic Fizeau drag effect[16–20], among many others. High quality factor surface plasmons in any two-dimensional material is key to the observation of these near-field plasmonic phenomena. However, the intrinsically high electronic losses of the plasmons is a significant barrier to their performance and applicability[21,22]. Hence, it is highly desirable to overcome the fundamental limitation on quality factor set by the electronic losses. One route to overcoming this limitation is to introduce enough gain to compensate for the intrinsic losses[23].

Gain, or wave amplification, is a phenomenon that is ubiquitous in nature. Rogue waves, Rijke tubes, and optical parametric amplification are all examples in which a wave is amplified rather than damped as it propagates through a medium[24–28]. In most of these examples, the amplification is driven by external pumping, such as a heater or laser

that can inject energy into the system. Early studies to impart gain to plasmons began with the realization that stimulated emission can also amplify plasmons in a way similar to amplification of light[29]. The first experimental demonstrations were able to amplify plasmons by coupling to external media that were either optically[30,31] or electrically[32] pumped, realizing plasmonic nanolasers with extremely compact field confinement. Optically pumped setups[33,34] require gain medium to be adjacent to the surface plasmon mode. Electrically pumped systems encapsulate a typical semiconductor laser system in conjunction with a plasmonic mode in a metal-insulator-metal gap, accompanied by population inversion within the insulating layer. However, integrating these approaches into on-chip nanophotonics has been challenging due to difficulties in down-sizing the optical gain components and issues related to contact resistance of small metallic structures[35,36].

Here, we propose that plasmons in a two-dimensional material with separated electron and hole pockets can experience internal plasmonic gain in the presence of an applied electrical current. The plasmonic medium derives its gain from nonlocal single particle transitions between the electron and hole bands and does not require

[1]Department of Electrical & Computer Engineering, University of Minnesota, Minneapolis, MN 55455, USA. [2]Department of Physics and Astronomy, University of Pennsylvania, Philadelphia, PA 19104, USA. ✉e-mail: tlow@umn.edu

an external gain medium. The electrons(holes) acquire a momentum anti-parallel(parallel) to the applied current direction. The shifted electron and hole distributions open up a finite energy momentum phase space window where population inversion can be achieved for single-particle transitions. When this phase space overlaps with the plasmon dispersion, plasmonic gain can occur. Such a condition for gain can be satisfied in two-dimensional materials with low carrier density when the electron and hole pockets have a separation on the order of the plasmon momentum. Examples of materials that meet these prerequisites are found in the 1T′-MX$_2$ family of transitional metal dichalcogenides such as WTe$_2$ and MoTe$_2$[37,38]. Although optical gain can also be induced by intraband processes, it is typically not coupled to the plasmon due to a mismatch in momentum and energy. Using WTe$_2$ provides a unique advantage over graphene because it has optical gain from interband processes that are coupled to the plasmon dispersion. Such a process does not exist in graphene. Furthermore, as schematically shown in Fig. 1, the transitions responsible for gain in our setup are non-reciprocal whereas conventional optically pumped systems are only capable of providing reciprocal gain. Coupled with the non-reciprocity of the plasmon dispersion itself, we are able to demonstrate that the amplified plasmons are highly non-reciprocal and can be channeled.

## Results

### The tilted Dirac model

The inverted band structure found in many 1T′-MX$_2$[37,38] materials follows the tilted 2D massive Dirac model[39,40]

$$H = t\eta k_x + v(k_y \sigma_x + k_x \sigma_y) + (m/2 - \alpha k^2)\sigma_z \quad (1)$$

where $t$ is the tilt along the $k_x$ direction, $\eta = \pm 1$ is the node index, $v$ is the Fermi velocity, $m$ is the gap size, and $\alpha$ is a term that closes the Fermi surface at large $k$. This minimal model captures most of the salient features of the unique inverted band structure. The energy dispersion of eq. (1) given by $\varepsilon_\pm(\mathbf{k}) = t\eta k_x \pm [v^2 k^2 + (m/2 - \alpha k^2)^2]^{1/2}$ shows an interesting dependence on the tilt angle parameterized by the ratio $t/v$ (see Fig. 2a). For $t/v < 1$ the model describes a type-I Dirac node which has a single electron or hole pocket at Fermi energies in the conduction or valence band. For $t/v > 1$ we find a type-II Dirac node which can support both an electron and hole Fermi surface where $\hat{\mathbf{k}}_{e-h}$, the relative displacement vector of the electron and hole pockets, is along the tilt direction. Here, we will be focusing on the type-II Dirac node with $E_F = 0$ eV which has electron and hole pockets of equal size. For definiteness, the parameters of the Dirac model are set to fit the tight-binding band structure of monolayer WTe$_2$[41] and are given as $v = 2.427$ eVÅ, $t = 1.16v$, $m = 0.138$ eV, $\alpha = 7.541$ eVÅ$^2$. Note that two

copies of the tilted Dirac model with $\eta = \pm 1$ are required to represent the two valleys in the WTe$_2$ band structure. A more detailed discussion of using eq. (1) to model the band structure of WTe$_2$ is given in the Supplementary information.

### Response function under current bias

Gain in the plasmons can be seen from a nonlocal description of the electron response. The plasmon excitations can be found by calculating the loss function $L(\mathbf{q},\omega) = -\text{Im}(1/\epsilon(\mathbf{q},\omega)) = \epsilon''/(\epsilon'^2 + \epsilon''^2)$ where $\epsilon = \epsilon' + i\epsilon''$. The plasmon solutions are given by $\epsilon(\mathbf{q},\omega) = 0$ which show up as a pole in the loss function. The sign of the loss function is given by $\epsilon''$ which is related to the loss or gain. A negative(positive) valued peak in the loss function therefore signifies a plasmon with gain(loss). A DC current bias $\mathbf{j}$ applied to the system may then be modeled by modifying the Fermi distribution to $f_u(E_\mathbf{k}) = \left(e^{(E_\mathbf{k} - \mathbf{u}\cdot\mathbf{k} - E_F)/k_B T} + 1\right)^{-1}$[42] where $\mathbf{u}$ is a parameter induced by the current density. The modified carrier occupation in both valleys for a $\mathbf{u}$ along the positive $x$-direction is shown in Fig. 2d,e. The applied electrical current requires that the electron(hole) pocket is shifted in the direction antiparallel(parallel) to the current. Since the relative orientation of the electron and hole pockets are opposite for the two valleys we find that they are shifted shifted closer to(away from) each other in the $\eta = +1(\eta = -1)$ valley.

The loss function for a drift current $u/v = 0.41$ and $\mathbf{q} = q_x\hat{\mathbf{x}}$ is shown in Fig. 2a. We first note that the plasmon dispersion is skewed in the direction opposite to the drift current and exhibits a very prominent non-reciprocal behavior. Moreover, the loss function in the $-q_x$ plasmon branch becomes negative signifying gain in the plasmon. The origin of gain can be understood by examining the imaginary part of the dielectric function from the $\eta = +1$ node shown in Fig. 2b. The bias creates two regions in which the imaginary part of the dielectric function is negative (blue regions labeled intra and inter in Fig. 2b). Both these regions originate from single particle transitions from a higher energy to lower energy, i.e. radiative relaxation of an inverted population. The intraband gain transitions occur mainly at $(\mathbf{q}, \omega)$ that are distant from the plasmonic spectrum. In contrast, the interband transitions between the electron and hole pockets occur at $(\mathbf{q}, \omega)$ that intersect directly with the plasmon dispersion. As the bias is increased, the interband gain region can overlap with the plasmonic spectrum and impart gain. We may confirm that the plasmon gain originates from interband transitions by examining the boundary for interband transitions (see dashed magenta line in Fig. 2b) which matches well with the onset of plasmonic gain as shown in Fig. 2b. The relation between plasmon gain and interband transitions implies that the gain window will depend on the electron-hole pocket separation. In the Supplementary information we show that the gain window is indeed shifted as the electron-hole pocket separation is tuned. However, as long as the gain window overlaps with the plasmon dispersion,

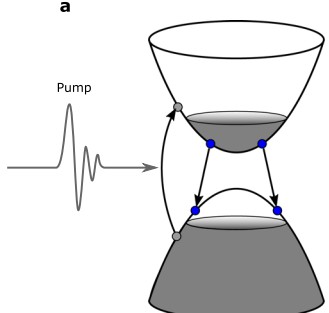

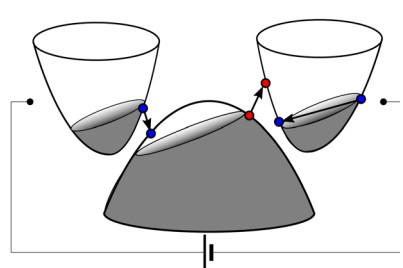

**Fig. 1 | Comparison of population inversion from a conventional optical pumping setup and a current biased tilted Dirac node. a** Schematic of population inversion achieved by a conventional optical pumping setup is shown. **b** Schematic of population inversion in current biased tilted Dirac nodes. The

blue(red) circles represent transitions that can contribute to emission(absorption) of a plasmon. Note how the transitions in the optically pumped case are reciprocal while for the tilted Dirac node they become non-reciprocal.

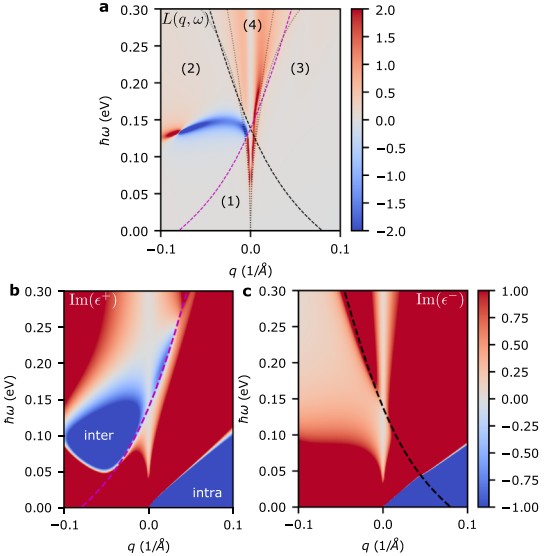

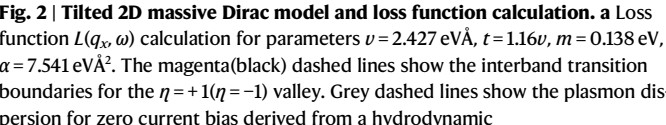

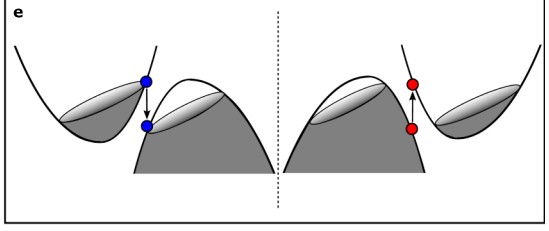

**Fig. 2 | Tilted 2D massive Dirac model and loss function calculation. a** Loss function $L(q_x, \omega)$ calculation for parameters $v = 2.427$ eVÅ, $t = 1.16v$, $m = 0.138$ eV, $\alpha = 7.541$ eVÅ$^2$. The magenta(black) dashed lines show the interband transition boundaries for the $\eta = +1(\eta = -1)$ valley. Grey dashed lines show the plasmon dispersion for zero current bias derived from a hydrodynamic model (see Supplementary Information). **b**, **c** Imaginary part of the dielectric function for the $\eta = +1$ and $\eta = -1$ valleys. The negative valued regions shown in blue are where the plasmon can experience gain. **d**, **e** Schematic representation of transitions in region (2) and (4) of Fig. 2a respectively.

the phenomena we discuss for the specific parameters of WTe$_2$ will remain unchanged.

Note that the plasmonic gain as shown by the loss function in Fig. 2a does not perfectly match the gain phase space suggested by the dielectric function in Fig. 2b. Specifically, the positive $q$ branch of the plasmon is not experiencing any gain despite overlapping with the interband gain phase space. This can be explained by taking into account the transitions from both the $\eta = 1$ and $\eta = -1$ valleys. Figure 2a has 4 regions that are separated by the interband transition boundaries of each valley. Region 1 has no interband transitions while regions 2 and 3 have transitions from $\eta = +1$ and $\eta = -1$ respectively. The plasmon branch that resides in region 2 therefore experiences the gain from single particle interband transitions in the $\eta = +1$ valley. Transitions in region 2 are schematically shown in Fig. 2d. Plasmons in region 4 are influenced by interband transitions from both valleys. Transitions from the $\eta = -1$ valley cancel out the gain from transitions in the $\eta = +1$ valley resulting in no net gain. This process is shown in Fig. 2e.

## Threshold current for plasmon gain

Having shown that plasmonic gain can indeed emerge for the tilted Dirac system, we now determine the threshold $u_{th}$ required for gain as a function of the model parameters. Since $\alpha$ closes the Fermi surface and therefore controls the size of the electron and hole pockets, we explore $u_{th}$ as a function of $\alpha$ while leaving all other parameters fixed to the values used for Fig. 2. A larger $\alpha$ will in general lead to a smaller Fermi pocket size. Let us first define the metric for quantifying the gain in the system as

$$g(\alpha, u) = |\int_{L<0} L(q, \omega) dq d\omega| \qquad (2)$$

where the integral is performed only over the region with gain ($L < 0$). Note that the gain metric defined is not the gain coefficient quantifying the exponential coefficient for amplification. Rather, $g(\alpha, u)$ is simply a measure of the total available gain in the system that allows us to capture two important features of the system. First, it accounts for any type of single particle transition that is capable of contributing to gain. More importantly, it allows us to infer when the plasmons begin to experience gain because the plasmons are represented as poles in the loss function and thus give a much larger contribution to $g(\alpha, u)$ than

single particle transitions. Results for $g(\alpha, u)$ are shown in Fig. 3a. A clear linear boundary is observed between regions with and without plasmonic gain (a detailed analysis of the linear phase boundary is given in the Supplementary information). Since the shift in the Fermi surface is proportional to both the parameter $u$ and the unbiased $k_F$, we can expect that a smaller Fermi surface will require a larger $u$ to reach the gain threshold. The threshold current density, which is dependent on both $u$ and the carrier density, is found to be on the order of a few mA$\mu$m$^{-1}$ and also decrease as a function of $\alpha$ (see SI). To take a closer look at what happens at the onset of gain, $g(\alpha, u)$ as a function of $u$ at $\alpha = 8$ eVÅ$^2$ is shown in Fig. 3b. For $u < u_{th}$, the slow increase in gain is due to intraband transitions as we can see from the loss function (inset). For $u > u_{th}$, there is a sharp increase in gain which originates from the onset of interband transitions imparting gain to the plasmon. Note the difference in scale between the two loss function insets which explains why such an abrupt transition is observed for $g(\alpha, u)$.

## Propagation of the amplified plasmon

We now expand our description of the plasmon to a general $\mathbf{q} = (q_x, q_y)$. At a given frequency, the complex plasmon wavevector $\mathbf{q}_{pl} = \mathbf{q}'_{pl} + i\mathbf{q}''_{pl}$ can be found by solving for the complex roots of $\epsilon(\mathbf{q}, \omega) = 0$. The isofrequency curves for $u/v = 0$ and $u/v = 0.41$ are shown in Fig. 4. As observed for the loss function in Fig. 2, a finite drift velocity breaks reciprocity and induces gain in the plasmon spectrum. The isofrequency curves shown in Fig. 4c,d further reveal that both the gain experienced by the plasmon for non-zero drift velocity and the topology of the isofrequency contour evolve with frequency. We find that at low frequency the isofrequency contour is separated into two disconnected closed contours, and that the amplified plasmons have a group velocity primarily along the vector connecting the electron and hole pockets $\hat{\mathbf{k}}_{e-h}$. As the frequency is increased above $\hbar\omega = 0.15$ eV, these pockets merge into a single closed isofrequency surface which is relatively 'flat' in the region of gain and has a group velocity perpendicular to $\hat{\mathbf{k}}_{e-h}$. Hence, highly directional propagation of the amplified plasmons perpendicular to $\hat{\mathbf{k}}_{e-h}$ is expected.

From the isofrequency plots we are able to deduce that the amplified plasmons will have a group velocity collimated perpendicular to $\hat{\mathbf{k}}_{e-h}$. To explicitly verify this behavior we now examine the real space propoagation of the plasmons launched by a point source. The

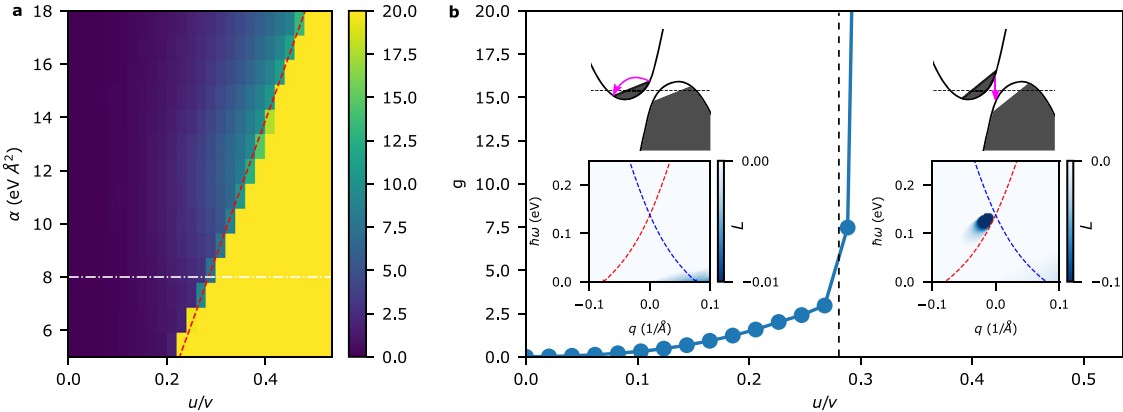

**Fig. 3 | Threshold drift velocity for gain. a** The gain metric $g(\alpha, u)$ is shown as a 2D color plot. A clear linear boundary between regions with and without plasmonic gain is observed. The red dashed line is a linear fitting to the transition boundary that gives $\alpha = 50.99\ u/v - 6.508$. **b** A slice of the 2D color plot at $\alpha = 8$ eVÅ² (white dashed line in Fig. 3a) is shown. The inset band structures schematically show the transitions responsible for $u/v = 0.165$ and $u/v = 0.33$ which are below and above $u_{th}$ respectively. The inset loss functions are scaled to only represent the regions over which $g(\alpha, u)$ is integrated as defined by $g(\alpha, u) = |\int_{L<0} L(q, \omega) dq d\omega|$. A black dashed line is drawn at $u_{th}/v = 0.285$.

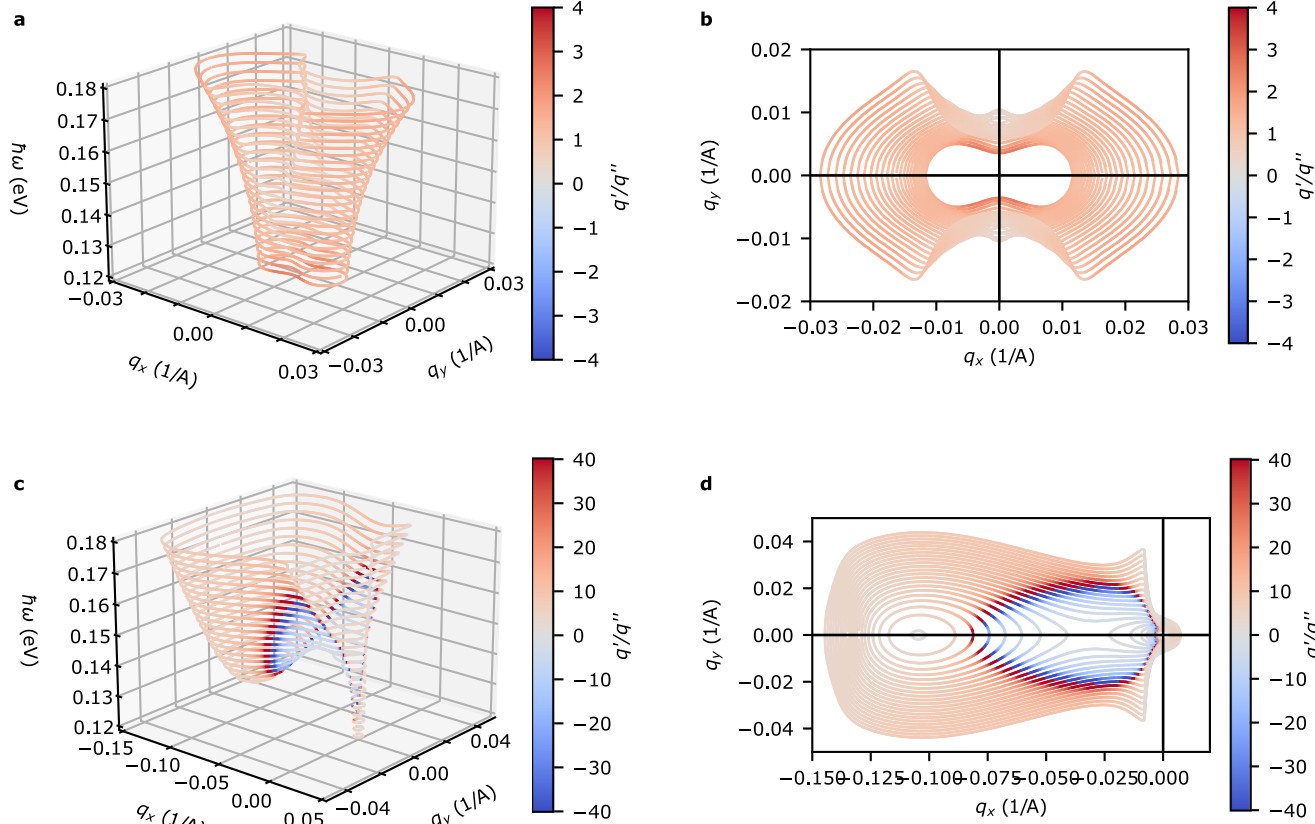

**Fig. 4 | Isofrequency curves for the plasmon dispersion. a** Isofrequency curves for $u = 0$. All other parameters for the Dirac model are identical to parameters used in Fig. 2. Color scale quantifies the quality factor $q'/q''$ which gives the number of cycles the plasmon goes through before decaying or amplifying by $e$. **c** Isofrequency curves for $u = 1$ eVÅ ($u/v = 0.41$). Negative values of the quality factor denote regions of the plasmon dispersion that experience gain. **b, d** Top view of the isofrequency curves shown in Fig. 4a, c. Arrow direction shows direction of increasing frequency.

plasmon field can be written as a superposition of unidirectional plane-waves

$$\mathbf{E}(\mathbf{r}) \propto \sum_{\mathbf{q}_{pl}} e^{i\mathbf{q}'_{pl} \cdot \mathbf{r}} e^{-\mathbf{q}''_{pl} \cdot \mathbf{r}} \Theta(\mathbf{v}_g \cdot \mathbf{r}). \tag{3}$$

$\Theta$ is a step function that describes a one-sided plane wave propagating in the direction of the group velocity. Results for the plane wave superposition for $\hbar\omega = 0.16$ eV with drift velocities $u < u_{th}$ and $u > u_{th}$ are shown in Fig. 5. For both cases, the plasmon propagation direction is roughly along the $y$-axis as expected from the isofrequency plots. It is important to emphasize that this is different from what one would

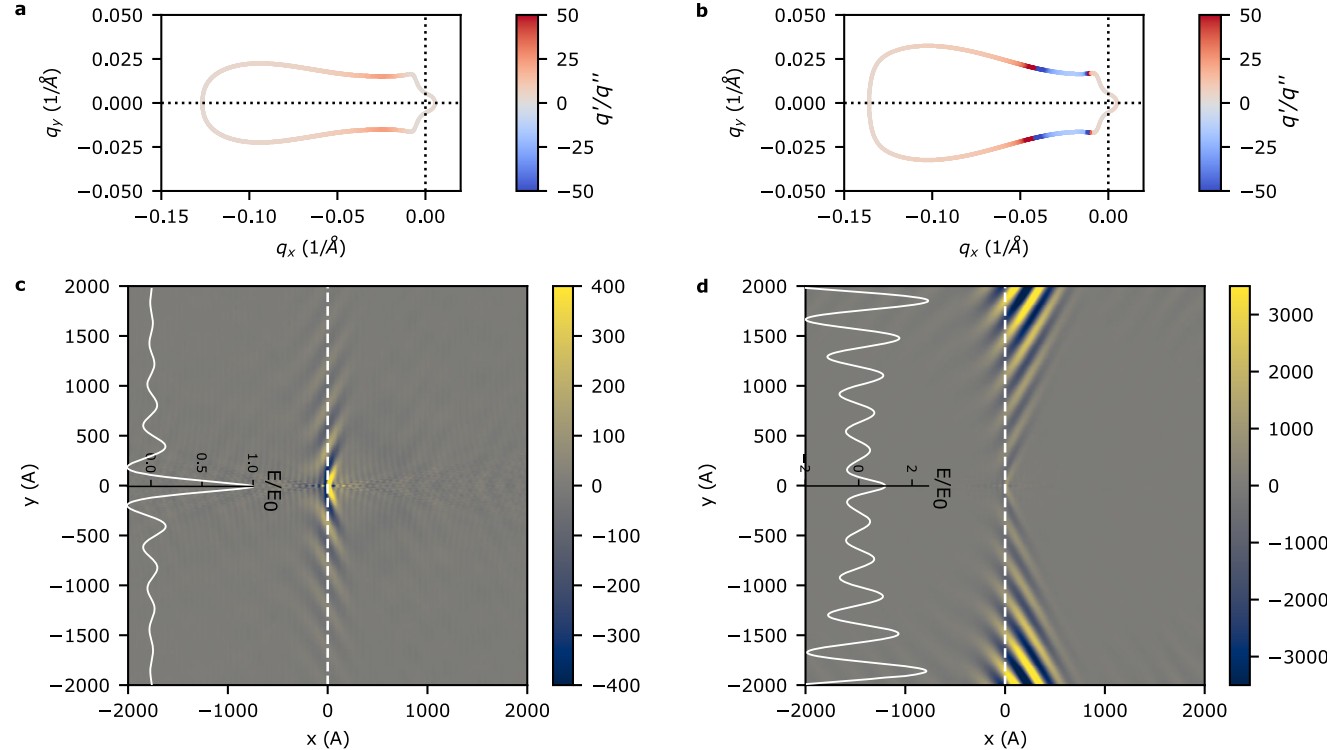

**Fig. 5 | Fields excited by a point dipole. a**, **b** Isofrequency curves at $\hbar\omega = 0.16$ eV for drift velocity $u/v = 0.37$ and $u/v = 0.41$ respectively. **c**, **d** Fields generated by eq. (3) for isofrequency curves Fig. 5a and b respectively. The solid white inset plots show the field amplitude along the white dashed lines and is normalized to the field at the origin.

expect from the phase velocity, which lies primarily in the $-q_x$-direction. For $u > u_{th}$, amplification of the plasmon in the direction of $\mathbf{v}_g$ is clearly observed.

## Discussion

In summary, we have demonstrated that plasmonic gain may be achieved by applying an electric current bias to materials with a type-II Dirac node band stucture. While we have used parameters that were fitted to the band structure of WTe$_2$ for most results presented in this work, the gain phenomena observed is generic and can be anticipated for semimetals with an electron-hole pocket separation on the order of the plasmon wavevector. General requirements for observing gain are as follows.

The material must have electron and hole pockets with a momentum space separation on the order of the plasmon wave vector and the current must be applied along the direction of separation. As a result, gain can be imparted to plasmons when its dispersion overlaps with the current driven population inverted single particle transition phase space. We have shown in Fig. 2 that for the type-II Dirac node, while both intra- and interband transitions allow for plasmon emission, the intraband transitions are well separated from the plasmon dispersion in the $(\mathbf{q}, \omega)$ space. The interband transitions on the other hand overlap with the plasmon dispersion. Also, recall that contributions from both the $\eta = +1$ and $\eta = -1$ valleys must be included. A given plasmon emission process $(\mathbf{q}, \omega)$ in valley $\eta = +1$ may correspond to plasmon absorption in valley $\eta = -1$ which overall cancels out the effect of gain to the plasmon. Hence the emission process that imparts gain to the plasmon must not have an absorbing counterpart in the opposite valley. When these conditions are met, we have shown that internal plasmonic gain may be imparted, thus providing a convenient route towards overcoming the intrinsic loss in the plasmon. The current density required to achieve gain is found to be on the order of a few mA$\mu m^{-1}$ which is achievable in monolayer graphene[43] and thin film WTe$_2$[44]. Furthermore, we find that the directionality of the plasmon

propagation becomes highly dependent on frequency and current magnitude. Thus our proposed setup presents a simple pathway for overcoming plasmonic loss for tightly confined two-dimensional plasmons while also providing control over their propagation direction. Propagation of these plasmons should be readily observable using a typical near-field scanning optical microscope setup[16].

## Methods
### Dielectric response calculation
The dielectric response is calculated from the random phase approximation to be $\epsilon(\mathbf{q}, \omega) = 1 - v_c \Pi(\mathbf{q}, \omega)$ where $v_c$ is the Coulomb potential and $\Pi(\mathbf{q}, \omega)$ is the non-interacting polarizability. The polarizability is given by a sum of the contributions from two oppositely tilted nodes ($\Pi = \Pi^+ + \Pi^-$) where the polarizability of each node is given by

$$\Pi^\eta(\mathbf{q},\omega) = \frac{g_s}{4\pi^2} \int d^2\mathbf{k} \sum_{ss'} \frac{f(E^\eta_{s\mathbf{k}}) - f(E^\eta_{s'\mathbf{k}'})}{E^\eta_{s\mathbf{k}} - E^\eta_{s'\mathbf{k}'} + \hbar\omega + i\delta} |\langle \psi^\eta_{s\mathbf{k}} | \psi^\eta_{s'\mathbf{k}'} \rangle|^2. \quad (4)$$

Here, $f(E_\mathbf{k})$ is the Fermi distribution, $\mathbf{k}' = \mathbf{k} + \mathbf{q}$, $E^\eta_{s\mathbf{k}}$ and $|\psi^\eta_{s\mathbf{k}}\rangle$ are the eigenvalues and eigenvectors of the tilted 2D massive Dirac Hamiltonian, and $g_s$ is the spin degeneracy. For all calculations, the Dirac model parameters are set to $v = 2.427$ eVÅ, $t = 1.16v$, $m = 0.138$ eV, $\alpha = 7.541$ eVÅ$^2$ and the temperature for the Fermi distribution is set to 5 meV.

Plasmon solutions are given by the complex roots of $\epsilon(\mathbf{q}, \omega) = 0$. At a given frequency $\omega$ and for small $\mathbf{q}''$, the complex roots are found by solving

$$\text{Re}\left[\epsilon(\mathbf{q}'_{pl})\right] = 0 \quad (5a)$$

$$\mathbf{q}''_{pl} \cdot \left[\nabla_{\mathbf{q}'} \text{Re}(\epsilon)\right]_{\mathbf{q}'_{pl}} = -\text{Im}\left[\epsilon(\mathbf{q}'_{pl})\right]. \quad (5b)$$

Note, however, that eq. (5b) does not uniquely define the direction of $\mathbf{q}_{pl}''$. $\mathbf{q}_{pl}''$ is taken to be parallel to the group velocity as the gain or loss is expected to be in the direction of the energy flow.

## Data availability

Data supporting key conclusions of this work are included within the article and Supplementary information. All raw data used in the current study are available from the corresponding author under reasonable request.

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

## Acknowledgements

All authors acknowledge support by the National Science Foundation, NSF/EFRI Grant No. EFRI-1741660. TL also acknowledges funding support from NSF DMREF under Grant Agreement No. 1921629.

## Author contributions

T.L. and E.M. conceived and supervised the project. S.H.P. and M.S. performed the numerical calculations and analysis. All authors discussed the results and wrote the paper.

## Competing interests
The authors declare no competing interests.
