## [Peer Review File · Nature Communications]

Reviewers' Comments:

Reviewer #1:

Remarks to the Author:

In this work, the authors propose an interesting strategy to generate plasmonic gain using the special electronic structure of current biased tilted Dirac nodes. They further support their proposal with the calculations of the plasmonic properties of 1T'-WTe₂. I think this work is quite interesting and novel and I recommend it for the publication in Nature Communications.

The text is general well-written. I have a couple comments that may help further improve the paper.

1. The authors calculate the polarizability by summing up the contributions from two oppositely tilted nodes instead of having a general equation to calculate all the contributions automatically. I wonder if they are doubly counting the contribution from the hole pocket as both tilted nodes share the same hole pocket. This may need to be clarified.

2. In the second paragraph on page 4, "...by the loss function in fig. 1a does not..." should be "...in fig. 1b..."

Reviewer #2:

Remarks to the Author:

The paper, "Plasmonic gain in current biased tilted Dirac nodes" reports the gain in plasmon modes of tilted type II Dirac materials with two kinds of charged fluids (electron and hole) in presence of longitudinal current. The effect of longitudinal current is introduced in the distribution function. In the presence of finite drift, the current density from electron and hole pockets does not cancel each other and gives rise to a finite contribution. The authors took low energy Dirac model with finite tilt and calculated the polarizability for electron and hole pockets separately. The authors argued that the plasmonic gain can occur when plasmonic dispersion overlaps with the current driven population inverted single-particle transition phase space.

In my view, the paper is too specialized for the general audience of Nature communication, and it is more suitable for specialized journals such as Phys. Rev. B. There are several existing works in the literature [Duppen et. al. 2016; Sabbaghi. et.al. 2015; Morgado 2018; Papaj 2020 etc] on current driven non-reciprocal plasmon. The authors adopted a similar mechanism for the tilted type II Dirac model of Ref. [38,39]. Having said that, I like the overall writing and presentation of the paper, though several questions remain for the general audience as well as for the experts.

Q1) One of the main points of that paper, is that the population inversion gives rise to a plasmonic gain. This idea is not made clear in the paper. What is the connection, will it happen in all systems where the plasmon frequencies lies in the energy window of population inversion and so on.

Q2) For the general audience, it should be clearly spelled out what does plasmonic gain mean. Why negative value of the Loss function corresponds to plasmonic gain? And why does the integral of the loss function, labeled $g(\alpha, \omega)$ in Eqn. 3, represent the gain coefficient?

Q3) Naively, I would think that the negative value of the imaginary part of the polarizability in Fig. 1 (c) signifies the single-particle excitation region, where the plasmon mode gets damped (via Landau damping). How are the authors getting gain in the plasmon spectrum within this particle-hole continuum boundary? This and several such questions should be made clear.

Q4) For the general audience, this may not be clear: "The applied electrical current requires that the electron and hole pockets in the $\eta = +1$ ($\eta = -1$) node are shifted closer to (away from) each other". Why do the electron and hole pockets shift differently for the two valleys?

Q5) In the definition of the quality factor q_1/q_2 in Figure-3, what do q_1 and q_2 represent? This should be elaborated in the main text.

Q6) The discussion related to the Fig.4 of the main text is not clear. What message do the authors

want to convey through Figure-4?

Q7) In this work, the momentum separation between the electron and hole pocket should be an important parameter and should be playing a crucial role. However, that is not discussed at all?

Q8) In the SM section on the hydrodynamic theory, the authors derive two plasmon modes for this system. However, none of the loss function plots seems to be showing this pair of split plasmon modes (which should be degenerate for small q)? If this pair of split plasmon modes is indeed there, then there should be discussions about the plasmonic gain for each of the modes.

Minor point: In the supplementary part S1, there is a typo in writing the power of effective mass (m instead of m^2) in all the four elements of the determinant.

Though I like the paper from a technical viewpoint, I cannot recommend this for publication in Nature Communication. It builds on several works in the literature, and highlights a new possibility. This new possibility of plasmonic gain is interesting, but not significant enough for Nature communication. The paper also needs to be improved significantly to highlight and clarify the physics being discussed even for a more technical journal.

Response to referees' comments

Sang Hyun Park,¹ Michael Sammon,¹ Eugene Mele,² and Tony Low^{1,*}

¹*Department of Electrical & Computer Engineering,
University of Minnesota, Minneapolis, Minnesota, 55455, USA*

²*Department of Physics and Astronomy, University of
Pennsylvania, Philadelphia, Pennsylvania, 19104, USA*

Manuscript ID: NCOMMS-22-17360-T

I. RESPONSE TO REVIEWER 1

In this work, the authors propose an interesting strategy to generate plasmonic gain using the special electronic structure of current biased tilted Dirac nodes. They further support their proposal with the calculations of the plasmonic properties of 1T-WTe₂. I think this work is quite interesting and novel and I recommend it for the publication in Nature Communications. The text is general well-written. I have a couple comments that may help further improve the paper.

We would like to thank the Referee for his/her supporting comments. Answers to the comments by the referee are provided below.

The authors calculate the polarizability by summing up the contributions from two oppositely tilted nodes instead of having a general equation to calculate all the contributions automatically. I wonder if they are doubly counting the contribution from the hole pocket as both tilted nodes share the same hole pocket. This may need to be clarified.

As pointed out by the referee, we are indeed double counting the hole pocket since there is one hole pocket per valley in the Dirac model while a single hole pocket is shared by both valleys in the actual WTe₂ band structure. However, when calculating the polarizability it is not the number of hole pockets but rather the number of total transitions we account for that is important. For interband transitions, consider a direct transition ($q = 0$) at a given energy $\hbar\omega$. The transition as

* tlow@umn.edu

FIG. R1. Contributions to polarizability calculation for (left) tilted Dirac model and (right) WTe_2 band structure.

described by using two copies of the tilted Dirac model and by using the bandstructure of WTe_2 are compared in fig. R1. It is easy to see that while the Dirac model includes two hole pockets, the number of direct transitions it accounts for is identical to the WTe_2 band structure. For intraband transitions in the valence band, note that the valence band given by the tilted Dirac model is asymmetric. Therefore an allowed transition at (q, ω) in the $\eta = +1$ node does not have a corresponding transition in the $\eta = -1$ node. Rather, since the band structures of the two nodes are mirror images of each other, for every transition (q, ω) in the $\eta = +1$ node, there will be a transition at $(-q, \omega)$ in the $\eta = -1$ node. This is exactly the required behavior for intraband transitions in the valence band for WTe_2 . This discussion is included to the supplementary information to avoid any confusion about the model we use to calculate the polarizability.

In the second paragraph on page 4, by the loss function in fig. 1a does not should be in fig. 1b

Thank you for pointing out this typo. The typo has been corrected in the revised version.

II. RESPONSE TO REVIEWER 2

In my view, the paper is too specialized for the general audience of Nature communication, and it is more suitable for specialized journals such as Phys. Rev. B. There are several existing works in the literature [Duppen et. al. 2016; Sabbaghi. et.al. 2015; Morgado 2018; Papaj 2020 etc] on current driven non-reciprocal plasmon. The authors adopted a similar mechanism for the tilted type II Dirac model of Ref. [38,39]. Having said that, I like the overall writing and presentation of the paper, though several questions remain for the general audience as well as for the experts.

We thank the reviewer for the constructive comments and are glad to hear that the writing and presentation was to their liking. Before addressing the questions presented by the referee, we would like to first emphasize the novelty of our work and explain in more detail how it differs from the existing works mentioned. The works mentioned by the referee focuses on plasmon non-reciprocity due to an applied current, and not plasmonic gain. To the best of our knowledge, our proposal for the generation of plasmonic gain via current is novel. In fact, this idea would not work in graphene. In current biased graphene, only intraband transitions will be able to provide gain and we have shown that the intraband gain phase space does not overlap with the plasmon dispersion. On the other hand, as we show in the manuscript, the closely located electron-hole pockets allow for gain via interband transitions that overlap with the plasmon dispersion. Furthermore, we would like to emphasize that previous work on plasmonic gain did not take into account the momentum dependence of stimulated emission transitions that are responsible for gain. In this work we show that the non-local population inversion window overlaps with the plasmon dispersion thus giving a description that incorporates the non-locality of the system. As a result we find that the both the plasmon and gain window become non-reciprocal.

Our approach offers a new approach to achieve population inversion. Most previous works discussing plasmon gain either optically pump an adjacent gain media or electrically pump a semiconductor lasing structure to achieve population inversion. Both setups require an external gain mechanism to be setup and pumped, after which the energy from stimulated emission is transferred to the plasmon system. However, our proposal of using type-II tilted Dirac nodes with a current bias is capable of supporting both the gain mechanism and the plasmon within a single material.

To illustrate the novelty of our work, we introduce a new figure (see fig. R2) that schematically shows the difference between our setup and previous optically pumped setups. Figure 1 of the original manuscript is also modified to schematically show the transitions that lead to gain. With this figure we first show that the transitions in the type-II Dirac node band structure that lead to plasmon gain are asymmetric in \mathbf{q} . This behavior is contrasted with the symmetric transitions given by an optically pumped setup. Furthermore, we are able to show the difference between intraband and interband gain transitions. The intraband gain in general does not overlap with the plasmon dispersion while the interband gain does. Finally, this schematic emphasizes that we are incorporating non-local transitions that lead to gain which is an important technical detail that has not been discussed before.

The following discussion has been added to the end of the introduction to highlight the novelty

FIG. R2. New figure 1 for manuscript. **a** Optically pumped population inversion and **b** current biased population inversion in type-II Dirac nodes are compared.

of our work. Although optical gain can also be induced by intraband processes, it is typically not coupled to the plasmon due to a mismatch in momentum and energy. Using WTe_2 provides a unique advantage over graphene because it has optical gain from interband processes that are coupled to the plasmon dispersion. Such a process does not exist in graphene. Furthermore, as schematically shown in fig. 1, the transitions responsible for gain in our setup are non-reciprocal whereas conventional optically pumped systems are only capable of providing reciprocal gain. Coupled with the non-reciprocity of the plasmon dispersion itself, we are able to demonstrate that the amplified plasmons are highly non-reciprocal and can be channeled.

One of the main points of that paper, is that the population inversion gives rise to a plasmonic gain. This idea is not made clear in the paper. What is the connection, will it happen in all systems where the plasmon frequencies lies in the energy window of population inversion and so on.

For the general audience, it should be clearly spelled out what does plasmonic gain mean. Why negative value of the Loss function corresponds to plasmonic gain? And why does the integral of the loss function, labeled $g(\alpha, \omega)$ in Eqn. 3, represent the gain coefficient?

Naively, I would think that the negative value of the imaginary part of the polarizability in Fig. 1 (c) signifies the single-particle excitation region, where the plasmon mode gets damped (via Landau damping). How are the authors getting gain in the plasmon spectrum within this particle-hole continuum boundary? This and several such questions should be made clear.

FIG. R3. Modified figure showing the **a** loss function, **b,c** imaginary part of the dielectric function, and **d,e** transitions in region (2) and (4) of **a** respectively.

The reviewer has several good questions about the polarizability and loss function calculations that we would like to clarify. We first note that there was an erroneous minus sign when plotting the imaginary part of the polarization. This error has been amended and applied to all plots of the dielectric function in the revised version of the manuscript. Signs for the loss function are not affected by this error.

The polarizability given in equation (2) represents the probability amplitude of generating an electron-hole pair with momentum q and energy $\hbar\omega$ in the non-interacting limit. Using the identity $\frac{1}{x+i\eta} = P(1/x) - i\pi\delta(x)$ the imaginary part of the polarizability may be written as

$$\text{Im}[\Pi^\eta(\mathbf{q}, \omega)] \propto - \int d^2\mathbf{k} \sum_{ss'} (f(E_{s\mathbf{k}}^\eta) - f(E_{s'\mathbf{k}'}^\eta)) \delta(E_{s\mathbf{k}}^\eta - E_{s'\mathbf{k}'}^\eta + \hbar\omega) |\langle \psi_{s\mathbf{k}}^\eta | \psi_{s'\mathbf{k}'}^\eta \rangle|^2.$$

From the delta function we see that the energy of state s', \mathbf{k}' must be higher than the energy of state s, \mathbf{k} since $\hbar\omega$ is a positive quantity. When the higher energy state is empty and the lower energy state is filled, $f(E_{s\mathbf{k}}^\eta) - f(E_{s'\mathbf{k}'}^\eta) = +1$ and thus $\text{Im}\Pi$ is negative. Landau damping is an example of such a process where the energy and momentum of a plasmon is absorbed to excite an electron-hole pair. When the higher energy state is full and lower energy state is empty $\text{Im}\Pi$ is positive, corresponding to the electron relaxing to lower energy and emitting energy. This can be considered the opposite process of Landau damping in which an electron-hole pair combine to emit a plasmon.

The interaction between plasmons and single-particle excitations described in the previous paragraph can be captured by calculating the RPA dielectric function and loss function. The plasmon dispersion is given by the zeros of $\epsilon(\mathbf{q}, \omega)$ which correspond to poles of the loss function. To interpret the sign of the loss function let us write

$$L = -\text{Im}(1/\epsilon) = \frac{\epsilon''}{\epsilon'^2 + \epsilon''^2}$$

where $\epsilon = \epsilon' + i\epsilon''$. In terms of the complex refractive index $\tilde{n} = n + i\kappa$ we have $\epsilon' = n^2 - \kappa^2$ and $\epsilon'' = 2n\kappa$. Since κ is directly connected to the amplification or attenuation of the plasmon we may also conclude that the sign of the loss function is a direct indication of plasmon loss or gain. A negative κ corresponds to amplification. Therefore a peak with negative values in the loss function indicates a plasmon experiencing amplification. The quantity $g(\alpha, u)$ that we have defined is not necessarily a gain coefficient but rather a simple measure of the total amount of gain available within a given range of \mathbf{q}, ω . Since plasmons appear as sharp peaks in the loss function, we are able to associate a abrupt jump in $g(\alpha, u)$ with the onset of plasmonic gain. This is exactly what we observe in figure 2 of the manuscript. The slow increase in g at low current densities originate from intraband population inversion that does not overlap with the plasmon dispersion. A sharp increase in g is then observed when the interband population inversion window overlaps with the plasmon dispersion resulting in plasmon amplification.

The loss function has been frequently used to show that plasmons experience damping when entering the particle-hole continuum boundary. However, the plasmon does not always have to experience damping when in the particle-hole continuum boundary. The transition boundaries that we draw in the loss function and polarizability plots correspond to the smallest $\hbar\omega$ at which the condition $\delta(E_{s\mathbf{k}}^\eta - E_{s'\mathbf{k}'}^\eta + \hbar\omega)$ can be satisfied but do not contain any information about the occupation of the states. We are able to observe gain in the particle-hole continuum because single-particle transitions in the gain window are high-to-low energy transitions (emission) and not low-to-high energy transitions (absorption). We can expect that whenever the plasmon dispersion overlaps with such a gain window, the plasmon will experience gain. For the zero bias case, there are no high-to-low energy transitions which is why we may always associate the particle-hole continuum with plasmon damping.

To make the relation between the single particle transitions and plasmonic gain clearer, we have modified figure 1 of the original manuscript to the figure shown in fig. R3. The loss function subplot is identical to the original manuscript. Subpanels b and c now show the imaginary part of the dielectric function for the two valleys which is easier to interpret in terms of loss and gain.

Subpanel d gives a schematic representation of transitions in region (2). In this region there is a range of (\mathbf{q}, ω) where plasmon-emitting transitions are available from the $\eta = +1$ valley while there are no absorbing transitions in the $\eta = -1$ which leads to net gain for the plasmon. Subpanel e is a representation of transitions in region (4). Here there are emitting transitions from $\eta = +1$ which can be cancelled out by absorbing transitions in $\eta = -1$. Therefore we do not observe any net gain for plasmon.

The following discussion has also been included to clarify the interpretation of the loss function. The plasmon excitations can be found by calculating the loss function $L(\mathbf{q}, \omega) = -\text{Im}(1/\epsilon(\mathbf{q}, \omega)) = \epsilon''/(\epsilon'^2 + \epsilon''^2)$ where $\epsilon = \epsilon' + i\epsilon''$. The plasmon solutions are given by $\epsilon(\mathbf{q}, \omega) = 0$ which show up as a pole in the loss function. The sign of the loss function is given by ϵ'' which is related to the loss or gain. A negative(positive) valued peak in the loss function therefore signifies a plasmon with gain(loss).

To clarify the role of the gain metric we have included the following. Note that the gain metric defined is not the gain coefficient quantifying the exponential coefficient for amplification. Rather, $g(\alpha, u)$ is simply a measure of the total available gain in the system that allows us to capture two important features of the system. First, it accounts for any type of single particle transition that is capable of contributing to gain. More importantly, it allows us to infer when the plasmons begin to experience gain because the plasmons are represented as poles in the loss function and thus give a much larger contribution to $g(\alpha, u)$ than single particle transitions.

For the general audience, this may not be clear: The applied electrical current requires that the electron and hole pockets in the $\eta = +1$ ($\eta = -1$) node are shifted closer to(away from) each other. Why do the electron and hole pockets shift differently for the two valleys?

The shift direction of the electron and hole pocket are actually identical for the two valleys. The electron pocket shifts in a direction opposite to the current whereas the hole pocket shifts in the direction of the current. However, the relative orientation of the electron and hole pockets are opposite for the two valleys. Hence we find that the electron and hole pockets are shifted closer to each other in the $\eta = +1$ valley while they shift away from each other in the $\eta = -1$ valley.

To clarify this point the manuscript has been changed as follows: The applied electrical current requires that the electron(hole) pocket is shifted in the direction antiparallel(parallel) to the current. Since the relative orientation of the electron and hole pockets are opposite for the two valleys we find that they are shifted shifted closer to(away from) each other in the $\eta = +1(\eta = -1)$ valley.

In the definition of the quality factor q_1/q_2 in Figure-3, what do q_1 and q_2 represents?

This should be elaborated in the main text.

Thank you for pointing this out. There has been a confusion of notation between the main text and figures. q_1 and q_2 in figure 3 each correspond to q'_{pl} and q''_{pl} in the main text, i.e. the real and imaginary parts of the complex plasmon wavevector. The notation in figure 3 has been changed to match the notation in the main text.

The discussion related to the Fig.4 of the main text is not clear. What message do the authors want to convey through Figure-4?

In principle, the plasmon dispersion as shown by the loss function and the isofrequency curves contain all information needed about the plasmon. However, it is still helpful to have a plot that explicitly shows the real space propagation of the plasmon rather than trying to deduce it from the dispersion relation. Figure 4 shows the real space propagation behavior for the plasmon below and above the gain threshold giving a clear visualization of the plasmon experiencing gain. More importantly, we also are able to see that the propagation direction of the plasmons are highly collimated. To make the message related to figure 4 clearer, we have added the following discussion to the manuscript. **From the isofrequency plots we are able to deduce that the amplified plasmons will have a group velocity collimated perpendicular to $\hat{\mathbf{k}}_{e-h}$. To explicitly verify this behavior we now examine the real space propoagation of the plasmons launched by a point source.**

In this work, the momentum separation between the electron and hole pocket should be an important parameter and should playing a crucial role. However, that is not discussed at all?

Indeed the momentum separation of the electron and hole pocket is an important parameter as it has an effect on the population inversion window. In terms of our model we are able to tune the electron-hole pocket separation using the mass parameter m . Calculating the polarizability of the $\eta = +1$ valley as a function of m reveals that the population inversion window is indeed shifted. However, this is a quantitative change that does not affect the general phenomena we are introducing. Nevertheless, since the electron-hole separation dependence is an important detail we have included an additional section to the supplementary information on these results. The following is added in the main text. **The relation between plasmon gain and interband transitions implies that the gain window will depend on the electron-hole pocket separation. In the supplementary information we show that the gain window is indeed shifted as the electron-hole pocket**

FIG. R4. Dielectric function at $q = -0.06 \text{ 1/\AA}$.

separation is tuned. However, as long as the gain window overlaps with the plasmon dispersion, the phenomena we discuss for the specific parameters of WTe_2 will remain unchanged.

In the SM section on the hydrodynamic theory, the authors derive two plasmon modes for this system. However, none of the loss function plots seems to be showing this pair of split plasmon modes (which should be degenerate for small q)? If this pair of split plasmon modes is indeed there, then there should be discussions about the plasmonic gain for each of the modes.

Thank you for pointing this out. Examining the dielectric function at higher energies, we indeed find an additional mode that corresponds to the split plasmon mode. However, this mode is at energies much higher than the gain window we are considering and hence is not studied in great detail. Nonetheless, to make this point clear we have elaborated the hydrodynamic theory section of the supplementary information to show that there are indeed two split modes, one of which is in the gain window and is therefore studied in more detail. fig. R4 shows the dielectric function at $q = -0.06 \text{ 1/\AA}$ from which we find a higher energy plasmon that is significantly damped.

Minor point: In the supplementary part S1, there is a typo in writing the power of effective mass (m instead of m^2) in all the four elements of the determinant.

Thank you for pointing out this typo. It has been fixed in the revised manuscript.

Reviewers' Comments:

Reviewer #2:

Remarks to the Author:

The authors have addressed all the concerns satisfactorily and significantly improved the manuscript's presentation.

The paper may be published in its current form.